# Effect of the Nonpathogenic Strain *Fusarium oxysporum* FO12 on Fe Acquisition in Rice (*Oryza sativa* L.) Plants

**DOI:** 10.3390/plants12173145

**Published:** 2023-08-31

**Authors:** Jorge Núñez-Cano, Francisco J. Romera, Pilar Prieto, María J. García, Jesús Sevillano-Caño, Carlos Agustí-Brisach, Rafael Pérez-Vicente, José Ramos, Carlos Lucena

**Affiliations:** 1Departamento de Agronomía (Unit of Excellence ‘María de Maeztu’ 2020-24), Edificio Celestino Mutis (C-4), Campus de Excelencia Internacional Agroalimentario de Rabanales (ceiA3), Universidad de Córdoba, 14071 Córdoba, Spain; jorgenunezcano@gmail.com (J.N.-C.); ag1roruf@uco.es (F.J.R.); b92gadem@uco.es (M.J.G.); o42secaj@uco.es (J.S.-C.); cagusti@uco.es (C.A.-B.); 2Departamento de Mejora Genética, Instituto de Agricultura Sostenible (IAS), Consejo Superior de Investigaciones Científicas (CSIC), 14004 Córdoba, Spain; pilar.prieto@ias.csic.es; 3Departamento de Botánica, Ecología y Fisiología Vegetal, Edificio Celestino Mutis (C-4), Campus de Excelencia Internacional Agroalimentario de Rabanales (ceiA3), Universidad de Córdoba, 14071 Córdoba, Spain; bv1pevir@uco.es; 4Departamento de Química Agrícola, Edafología y Microbiología, Edificio Severo Ochoa (C-6), Campus de Excelencia Internacional Agroalimentario de Rabanales (ceiA3), Universidad de Córdoba, 14071 Córdoba, Spain; mi1raruj@uco.es

**Keywords:** biostimulant, Fe deficiency, phytosiderophores, rhizosphere microorganisms, graminaceous plants

## Abstract

Rice (*Oryza sativa* L.) is a very important cereal worldwide, since it is the staple food for more than half of the world’s population. Iron (Fe) deficiency is among the most important agronomical concerns in calcareous soils where rice plants may suffer from this deficiency. Current production systems are based on the use of high-yielding varieties and the application of large quantities of agrochemicals, which can cause major environmental problems. The use of beneficial rhizosphere microorganisms is considered a relevant sustainable alternative to synthetic fertilizers. The main goal of this study was to determine the ability of the nonpathogenic strain *Fusarium oxysporum* FO12 to induce Fe-deficiency responses in rice plants and its effects on plant growth and Fe chlorosis. Experiments were carried out under hydroponic system conditions. Our results show that the root inoculation of rice plants with FO12 promotes the production of phytosiderophores and plant growth while reducing Fe chlorosis symptoms after several days of cultivation. Moreover, Fe-related genes are upregulated by FO12 at certain times in inoculated plants regardless of Fe conditions. This microorganism also colonizes root cortical tissues. In conclusion, FO12 enhances Fe-deficiency responses in rice plants, achieves growth promotion, and reduces Fe chlorosis symptoms.

## 1. Introduction

It is estimated that the world population will reach approximately 9 billion inhabitants by the year 2050, with an increase in food demand of 70% [1]. Rice cultivation is a very important cereal throughout the world, since it is the staple food for more than half of the world’s population. It is cultivated in more than 100 countries and provides more than 20% of the calories consumed worldwide [2]. Currently, production systems are mainly based on the use of high-yield varieties and the application of large amounts of agrochemicals, which leads to unsustainable agriculture [3]. Among the problems caused by these production systems, there is soil and groundwater contamination, imbalance of soil nutrients and reduction of soil biodiversity [4,5]. Given this situation, it is necessary to change to a sustainable production system which is more environmentally friendly and has less dependence on chemical fertilizers [6,7]. For this reason, the development of crop varieties more efficient in nutrient acquisition and a better management of the rhizosphere are necessary [8]. The rhizosphere is the fraction of the soil close to the roots, rich in energy and in which a large number of microbes, like rhizobacteria and fungi, live [9]. Some of these microbes associated with plants could be exploited to achieve promotion of growth and greater plant productivity [10]. Many of these mutualistic microbes release nutrient solubilizing compounds or modify the physiology and architecture of the roots in order to help plants to obtain nutrients like iron (Fe) and others [11,12,13,14,15].

Iron (Fe) is one of the most abundant elements in the earth’s lithosphere, but its solubility and availability for plants is low in calcareous soils with pH ranging from 7.4 to 8.5 [16,17]. In this case, plants may suffer from Fe deficiency, showing chlorosis in the youngest leaves [18,19,20]. Fe is a redox-active metal that is involved in hemoproteins related to electron transfer in photosynthesis and mitochondrial respiration, as well as protection against reactive oxygen species [21]. This nutrient is also involved in other key processes in plant physiology, such as chlorophyll biosynthesis, nitrogen assimilation and the biosynthesis of hormones like gibberellic acid, ethylene and jasmonic acid [21,22,23].

Plants have evolved two distinct strategies, namely Strategy I and Strategy II, to facilitate the uptake of Fe from the soil. Strategy I is employed by non-graminaceous plants, such as dicots, and involves the reduction of Fe^3+^ to Fe^2+^ before absorption [24,25]. This reduction process is facilitated by a ferric reductase located at the root surface, which is encoded by the *AtFRO2* gene in *Arabidopsis thaliana*. Subsequently, Fe^2+^ is taken up through an Fe^2+^ transporter, which is encoded by the *AtIRT1* gene in *Arabidopsis thaliana* [24,25]. When facing Fe deficiency, these Strategy I plants activate several physiological and morphological responses in their roots. These responses include an increased ferric reductase activity, enhanced capacity for Fe^2+^ uptake, acidification of the rhizosphere (due to H^+^-ATPases encoded by *AtAHA* genes in *Arabidopsis*), as well as escalated synthesis and release of organic acids (e.g., malate and citrate) and phenolic compounds (such as coumarins and others) [17,19]. Morphologically, noteworthy adaptations include the formation of subapical root hairs, cluster roots, and transfer cells, all aimed at increasing the root’s contact surface with the soil. The enhancement of both morphological and physiological responses is particularly significant in the subapical region of the roots [19,26].

To obtain Fe from the soil, Strategy II plant species release PhytoSiderophores (PS) from their roots, through transporters like the one encoded by the *TOM1* gene in rice, which form stable Fe^3+^ chelates with Fe^3+^ ions in the soil [27]. These Fe^3+^ chelates (Fe^3+^-PS) are then taken up by specific epidermal root cell plasma membrane transporters, like the one encoded by the *YSL15* gene in rice [25,28,29]. Under Fe-deficient conditions, Strategy II species greatly increase the production and release of PSand the number of Fe^3+^-PS transporters, and develop other physiological responses [29]. The increased production of PS is related to a higher expression of genes encoding transcription factors, like *IRO2*, which upregulate the expression of genes implicated in PS synthesis, like the *NAAT* gene, encoding the enzyme nicotianamine aminotransferase [30]. Rice, traditionally considered a Strategy II species [29], also presents some characteristics of Strategy I species, such as an enhanced Fe^2+^ uptake through a Fe^2+^ transporter, encoded by the *OsIRT1* gene [27,31,32]. For this reason, some authors consider it a plant species that uses a combined strategy [33,34,35].

Our group has demonstrated a role for ethylene, whose production increases in Fe-deficient roots, in the regulation of Fe-deficiency responses by Strategy I plant species [25,36]. However, there are few publications relating ethylene to Fe deficiency responses in Strategy II plant species [19,25]. In fact, Romera et al. [37] found that there were no differences in ethylene production between Fe-sufficient and Fe-deficient roots of several Strategy II plant species, like maize, wheat and barley. However, in the roots of rice plants that possesses a combined strategy, ethylene production is also higher under Fe-deficient conditions [38]. These results are reinforced by the discovery that, under Fe deficiency, ethylene synthesis genes, such as *OsACS*, *OsACO*, *OsSAMS* and *OsMTK*, are upregulated in rice roots [29,38,39]. *SAMS* and *MTK* genes are also involved in nicotianamine (NA) and PS synthesis [40,41].

It has been well demonstrated that some beneficial rhizosphere microorganisms, i.e., bacteria and fungi, are able to enhance plant nutrition and growth. These kinds of beneficial microorganisms are called Plant-Growth-Promoting Bacteria or Plant-Growth-Promoting Rhizobacteria (PGPB or PGPR, respectively) and Plant-Growth-Promoting Fungi (PGPF) [9]. Some of them can also boost plant defenses, rendering the entire plant more resistant to pathogens and pests, through a phenomenon called Induced Systemic Resistance (ISR) [9]. Some of the ISR-eliciting microorganisms are also able to enhance plant Fe nutrition by inducing Fe-deficiency responses because both processes (ISR and the responses) are modulated by similar hormones and signaling molecules, like ethylene and NO [19]. In Strategy I plants, the effects of these microorganisms on Fe nutrition are associated with their capacity to upregulate many key Fe-related genes, like *FIT*, *MYB72*, *IRT1*, *FRO2*, and others [9,19]. Nonetheless, their effects on Strategy II plants have been less studied [19].

In some cases, nonpathogenic strains of *Fusarium oxysporum* have been found to trigger Induced Systemic Resistance (ISR) [42,43], providing protection against soilborne pathogens like *Fusarium* spp. wilt and *Verticillium dahliae*-induced wilt, effectively reducing disease symptoms [42,43,44]. However, it has been proposed that these nonpathogenic strains might also induce a different type of resistance known as Endophytic Mediated Resistance (EMR). EMR occurs when a plant gains resistance against pathogens after being colonized by an endophytic microorganism, such as *Fusarium* spp. [45].

This form of resistance (EMR), unlike ISR, is characterized by endophytic microorganisms that typically do not provide protection against pathogens in the above-ground tissues [46]. Moreover, some studies suggest that ethylene does not play a role in this type of resistance [47]. However, the claim that nonpathogenic strains of *Fusarium oxysporum* induce this resistance (EMR) is a subject of debate due to conflicting evidence. For instance, research has shown that when *Capsicum annuum* plants are inoculated with the nonpathogenic strain of *F. oxysporum* FO47, they gain resistance against *Verticillium dahliae* and experience reduced foliar damage [48]. Constantin et al. [47] have proposed that endophytic microorganisms can induce EMR, a resistance mechanism distinct from ISR, with the intriguing feature of ethylene independence. This assertion implies that the introduction of nonpathogenic strains of F. oxysporum to plant hosts would activate resistance pathways not reliant on ethylene signaling. However, our own research, documented and published by our group, has revealed a contrary finding. Specifically, the inoculation of plants with the FO12 strain resulted in a noticeable augmentation of ethylene-related gene expression at specific intervals [49].

Within this framework, most recently, the nonpathogenic strain *F. oxysporum* FO12 has been characterized not only as a potential biological control agent of Verticillium wilt of olive, a disease caused by the soilborne pathogen *Verticllium dahliae* [44,50,51,52,53], but also as a resistance host inducer modulating in parallel the Fe acquisition in Arabidopsis and cucumber plant models [49]. In order to go ahead in generating knowledge on this last evidence, in this study, we have conducted experiments with rice plants (*Oryza sativa* L. cv. Puntal) using the nonpathogenic strain FO12 of *F. oxysporum* to demonstrate whether FO12 could induce Fe-deficiency responses in rice, as other ISR-eliciting fungi do in dicot plants [19].

## 2. Results

### 2.1. Effect of the Inoculation with FO12 on Fe Chlorosis

The main symptom of Fe deficiency in plants is an interveinal yellowing of the youngest leaves, known as Fe chlorosis. It is produced because Fe plays an important role in the functioning of some enzymes involved in the synthesis of chlorophyll [23,54]. Rice plants cultivated with Fe presented higher SPAD values than those without Fe (Figure 1). The inoculation caused a clear promotive effect on the SPAD index of plants grown under Fe deficiency but had almost no effect in those grown under Fe sufficiency (Figure 1).

### 2.2. Effect of the Inoculation with FO12 on Growth Promotion

To determine growth promotion, half of the 22 d old rice plants were inoculated with the FO12 strain, and then both inoculated and control plants were cultivated for 12 additional days, either under Fe deficiency (–Fe) or Fe 70 μM (+Fe). After 6 d of the inoculation with FO12, there was a significant growth-promoting effect both in shoots and roots just under Fe sufficiency (Figure 2a,b). However, no changes were observed both in shoots and roots relative to inoculation in Fe-deficient plants (Figure 2a,b).

### 2.3. Effect of the Inoculation with FO12 on Phytosiderophores Production

Regarding PS production, an increasing trend is observed up to 24 h, then it decreases substantially (Figure 3). In the first sampling carried out at 6 h, there are statistical differences between the treatments without Fe. The highest average value is observed in the treatment that was inoculated with FO12. In the case of treatments with Fe, differences between treatments are also observed. However, the plants that were inoculated with FO12 showed a lower average than those that were not inoculated. At 12 h, the plants of the treatments without Fe and inoculated with FO12 presented the highest mean PS production among all the treatments. Likewise, it was the only treatment where statistical differences were observed. The sampling carried out at 24 h was the one where the highest values were obtained in terms of PS production. The plants of the treatments without Fe plus FO12 had the highest means of this sampling. In the same way, statistical differences were observed both in the treatments with and without Fe, the plants of the treatments inoculated with FO12 being the ones that presented the highest values. At 48 h, PS production by the plants declined in all treatments (Figure 3).

### 2.4. Effect of the Inoculation with FO12 on Expression of Fe-Related Genes

The relative expressions of PS-related genes (*TOM1*, *IRO2*, *NAAT* and *YSL15*) by the roots of the rice plants, as well as that of the *IRT1* gene, were also analysed (Figure 4). Inoculation with FO12 strain had a clear inductor effect over the expression of all genes analysed in rice plants grown under Fe-deficient conditions. Similarly, most of the genes, except *NAAT* and *IRT1*, presented higher expression in the +Fe+FO12 treatment in relation to the +Fe treatment (control plants) (Figure 4).

### 2.5. Iron Concentration in the Plant Substrate Affects FO12 Colonization in Rice Root Tissues

With the aim of assessing the effect of Fe on the colonization process of rice root tissues by FO12, the GFP-tagged *F. oxysporum* was visualized using CLSM in the absence of Fe and compared with the colonization process observed in the presence of Fe at 70 μM concentration. Using the GFP-FO12 in CLSM experiments allowed the in situ visualization of the fungus on/in rice roots without any tissue manipulation. Rice roots were already colonized by conidia of *F. oxysporum* 1 dai (days after inoculation) in both treatments. Conidia were observed already germinating at this time point and did germinate similarly over rice roots in both treatments (–Fe and +Fe). After 4 dai, differences were already observed in GFP-FO12 colonization progress on rice roots, with the colonization of the rice root surface being more profuse in the absence of Fe than in the presence of Fe (Figure 5a,b).

During the progression of the experiment, GFP-FO12 developed on the root surface at a later stage (10 dai) both in the presence and in the absence of Fe (Figure 5c,d). Fungus conidia showed a slight reduction in the development of hyphae on the root surface of plants growing in the presence of Fe at 15 dai (Figure 5e,f). From this time point, we kept analysing plants by CLSM until 21 dai, although no substantial variations in plants were detected from 10 dai until the end of the experiment. Thus, GFP-FO12 reached a similar degree of hyphae development on the root surface of rice plants from 10 dai until the end of the bioassay. In fact, the visualization of the fungus at 18 dai (Figure 5g,h) showed an equivalent colonization degree to those of the preceding and following days until the end of the experiment (21 dai, Figure 5i,j). As with the observations taken previously, the proportion of hyphae were slightly lower in the root surface of plants growing with Fe at the end of the experiment (Figure 5i,j).

The colonization progress of GFP-FO12 on the rice root surface displayed non-uniformity, both in the presence and absence of Fe. Some regions of the roots exhibited a significant abundance of hyphal colonization, while other areas remained entirely devoid of the fungus. Utilizing Confocal Laser Scanning Microscopy (CLSM), we were able to identify internal colonization of rice roots by GFP-FO12. This internal colonization by GFP-FO12 was evident in the cortical tissue of rice roots starting from the early days of the bioassay and continued until the end, particularly in plants growing without Fe (Figure 5e and inset).

It did not detect GFP-FO12 hyphae proliferating in the vascular tissue at any time. In contrast, GFP-FO12 was only detected on the rice root surface up to the end of the experiment in the roots of rice plants growing with Fe, although we cannot rule out that internal colonization can also occur in plants in the presence of Fe.

Most of the conidia germinated and the proliferation of hyphae could be observed on the root surface in plants growing both in the presence and in the absence of Fe. A slightly higher proliferation of hyphae could be observed on the root surface in plants growing with no concentration of Fe, in the absence of iron from the beginning of the bioassay.

## 3. Discussion

To obtain Fe from the soil, Strategy II plant species release PhytoSiderophores (PS) from their roots, which form stable Fe^3+^ chelates (Fe^3+^-PS). PS are released through transporters, like the one encoded by the *TOM1* gene in rice, while the Fe^3+^-PS are taken up by transporters, like the one encoded by the *YSL15* gene in rice [25,27,28,29]. Under Fe-deficient conditions, Strategy II species greatly increase the production and release of PS and the number of Fe^3+^-PS transporters, and develop other physiological and regulatory responses [29]. Rice, traditionally considered a Strategy II species [29], also presents some characteristics of Strategy I species, such as an enhanced Fe^2+^ uptake through a Fe^2+^ transporter, encoded by the *OsIRT1* gene [27,31,32]. For this reason, some authors consider it a plant species that uses a combined strategy [33,34,35]. Despite the activation of these Fe-deficiency responses, crops can need an added contribution of fertilizers to withstand the enormous pressure that current agriculture exerts on crop production.

Searching for production strategies focused on gradually reducing the dependence on the application of large amounts of chemical products is one of the main challenges of current agriculture. The use of beneficial microorganisms is one of the strategies that is becoming more and more established every day. Microorganisms contribute to enhancing the tolerance of plants to abiotic stresses and to increasing their resistance to pathogens. In the same way, they can promote plant growth and increase the acquisition of nutrients through different mechanisms, such as changes in the soil structure and nutrient solubility, and changes in root morphology and physiology [55,56,57,58].

The results of this study show some positive effect of FO12 on plant growth and Fe acquisition by rice plants. The primary indication of Fe deficiency in plants manifests as interveinal yellowing in the most juvenile leaves, a condition referred to as Fe chlorosis. This discoloration arises due to iron’s pivotal function within several enzymes engaged in chlorophyll synthesis [23,54]. FO12 can affect the photosynthetic activity of the plants since those growing without Fe that were inoculated presented higher SPAD values (Figure 1). These results are similar to those obtained with inoculation with the fungus *Trichoderma asperellum* SL2 in rice [59]. However, the SPAD values did not show significant differences between control and FO12 inoculated treatments in cucumber plants cultured in a calcareous soil [49].

The effect on the development of the plants was reflected in a greater weight of both the aerial part and the roots (Figure 2). These positive effects have also been identified in various studies with the use of endophytic microbes [60,61,62]. FO12, in addition to inducing Fe-deficiency responses, exhibits growth-promoting properties, much like other Plant-Growth-Promoting Fungi (PGPF) and Plant-Growth-Promoting Bacteria (PGPB) [9]. Numerous studies have reported similar findings with other beneficial rhizosphere microorganisms. For instance, [63] utilized a combination of four microorganisms to inoculate soybean seeds, resulting in a noticeable growth-promoting effect on the plants. Similarly, Fontenelle et al. [64] observed a highly significant growth promotion when various isolates of *Trichoderma* spp. were applied to tomato plants under greenhouse conditions.

The application of PGPR or PGPF has proven to be an effective method in improving Fe chlorosis in calcareous soils [65,66,67]. Liu et al. [66] demonstrated the promotion of growth and enhanced mineral uptake in strawberry plants by inoculating them with isolates like *Agrobacterium*, *Bacillus* and *Alcaligenes*. Likewise, Liu et al. [66] achieved growth promotion in alfalfa plants by inoculating them with *Pseudomonas aeruginosa* and *Enterobacter aerogenes* cultured under saline–alkali conditions in a greenhouse. Moreover, El_Komy et al. [67] ameliorated the symptoms caused by *Fusarium solani*, *Macrophomina phaseolina* and *Rhizoctonia solani* in sunflower plants cultivated in a calcareous soil under field conditions, showing a clear growth-promoting effect with the inoculation of a mixture of rhizobacteria.

Many microbes produce signals that induce in the plant PS production and hormones that favor Fe acquisition [68]. In this study, the results obtained show that FO12 can induce a greater PS production by rice plants (Figure 3). This positive effect was higher when plants were growing without Fe (Figure 3). In order to summarize our results and the results of other authors [19,49] it seems that FO12 induces Fe-deficiency responses like other ISR-eliciting microorganisms.

According to Constantin et al. [47], endophytic micro-organisms induce EMR. These authors suggested that this type of resistance is independent of ethylene, in contrast to ISR. This statement means that the colonization of the plant by nonpathogenic strains of *F. oxysporum* would induce resistance in an ethylene-independent manner. However, Aparicio et al. [49] showed that the expression of ethylene-related genes was evidently enhanced at certain times under inoculation with FO12 in cucumber plants. Additionally, NO levels were also increased with the inoculation in the subapical region of the roots. Kavroulakis et al. [69] demonstrated that ethylene-deficient mutants of tomato inoculated with *F. solani* were more susceptible to pathogen attack, supporting a role for ethylene in the acquisition of resistance against *Fusarium*. Furthermore, NO and ethylene enhance the expression of several Fe acquisition genes in *Arabidopsis thaliana* [70,71]. Given that FO12 has been shown to upregulate ethylene-related genes and enhance NO production [49], it is plausible to consider that FO12 might induce iron deficiency responses in an ethylene/NO-dependent manner. Consequently, this implies that the FO12 strain could trigger Induced Systemic Resistance (ISR) rather than Endophytic-Mediated Resistance (EMR). Something similar could happen with rice plants, traditionally considered a Strategy II species but that possesses some characteristics of Strategy I species, in which several authors have shown that ethylene can also play a role in the regulation of some of its Fe-deficiency responses [25].

The expression of Fe-related genes in rice plants was evaluated (Figure 4). Results indicated a more enhanced expression in the presence of FO12. In cucumber plants, the FO12 strain induces the upregulation of Fe-related genes, including *CsFRO1*, *CsIRT1*, and *CsHA1* [49]. Similar effects have been observed in *Arabidopsis thaliana* when root-inoculated with the WCS417 strain of *Pseudomonas simiae*, leading to the increased expression of *MYB72*, *FRO2* and *IRT1* [11,72]. Moreover, the *Paenibacillus polymyxa* BFKC01 strain was found to promote growth and enhance the expression of *FRO2*, *FIT* and *IRT1* in *A. thaliana* [73]. Additionally, exposure of tomato plants to *Trichoderma* volatiles induced the expression of Fe-deficiency genes, such as *LeFRO*, *LeIRT*, and *LeFER* [74].

*Fusarium oxysporum*, being an endophytic microorganism, can grow inside the plant but typically does not colonize the vascular system. Instead, it forms fungal hyphae along the root cortex and endodermis [45]. Our findings align with this understanding, as the FO12 strain was observed to colonize the intercellular spaces of the cortical cells (Figure 5), supporting its endophytic nature. Notably, the inoculation of rice plants with the FO12 strain is facilitated by depleting Fe from the nutrient solution. On the other hand, the cortical tissue of rice roots was internally colonized by GFP-FO12 from early days until the end of the bioassay when plants were growing without Fe (Figure 5e, inset). These results agree with those obtained by Guerra & Anderson [75], showing that *Phaseolus vulgaris* plants growing in a hydroponic system with restricted Fe and B were more susceptible against *Fusarium* wilt. Similar results were obtained recently by Aparicio et al. [49], analyzing the time-course colonization processes of cucumber roots by the FO12-GFP-tagged strain. Collectively, these findings suggest that the depletion of Fe in the plant promotes endophytic colonization by non-pathogenic strains, such as FO12.

## 4. Materials and Methods

### 4.1. Seed Germination and Plant Cultivation

Experiments were carried out with rice (*Oriza sativa* L. var. ‘Puntal’) plants. Seeds were surface sterilized as described by Aparicio et al. [49]. Then, seedlings were transferred to a hydroponic system. Each of eight seedling groups were inserted in plastic lids and held in the holes of a thin polyurethane raft floating on an aerated nutrient solution containing 2 mM Ca(NO_3_)_2_, 0.75 mM K_2_SO_4_, 0.65 mM MgSO_4_, 0.5 mM KH_2_PO_4_, 50 µM KCl, 10 µM H_3_BO_3_, 1 µM MnSO_4_, 0.5 µM CuSO_4_, 0.5 µM ZnSO_4_, 0.05 µM (NH_4_)_6_Mo_7_O_24_ and 45 µM Fe-EDTA. When the plants were 22 d old, different treatments were applied: (1) +Fe (complete nutrient solution with 70 µM Fe-EDTA); (2) +Fe +FO12 (+Fe treatment + FO12 inoculum); (3) –Fe (nutrient solution without Fe-EDTA); and (4) –Fe +FO12 (–Fe treatment + FO12 inoculum). Plants were maintained in the treatments from 4 to 21 d, depending on the experiments.

### 4.2. Cultivation of Fungus and Inoculum Preparation

The FO12 strain and GFP-FO12, which was transformed with GFP (Green Fluorescent Protein) [76], were generously provided by the ‘Patología Agroforestal’ group from the ‘Universidad de Córdoba’. The FO12 strain was cultured in 250 mL of Potato Dextrose Broth (PDB, Scharlau) in 1-L flasks. Following a 4-day incubation period at 28 °C with continuous shaking at 110 rpm, the culture was filtered using a sterile nylon filter with a pore size of 10 µM (NY-LON_10, *Filtra Vibración*) to separate the spores from the mycelia. The spores were then centrifuged at 10,000 rpm for 10 min and subsequently resuspended in 4 mL of sterile distilled water. Prior to inoculation, the spore concentration was determined by counting them in a Neubauer chamber. As for GFP-FO12, the same culture method was employed, with the addition of 25 ppm of Hygromycin to maintain selective pressure.

### 4.3. Plant Inoculation

Following the method by Navarro-Velasco et al. [77], 22-day-old rice plants underwent root immersion for inoculation. Initially, a solution containing 10^7^ spores/mL was prepared using distilled water and poured into a 1.5 L capacity tray. To prevent contact between the solution and shoots, certain lanes were covered with insulating tape. The roots of the plants were then submerged in the solution for a duration of 30 min. Throughout the process, constant gentle shaking of the system was maintained to prevent spore precipitation. After inoculation, the plants were subsequently subjected to the mentioned treatments: +Fe or –Fe. For each treatment, there was a corresponding control group that was not inoculated.

### 4.4. Physiological and Morphological Assessments

Plants were periodically harvested (at 4, 8, 12 and 16 d after treatments) to observe growth promotion, by determining shoot height and both shoot and root fresh weights. Furthermore, in this study, Fe chlorosis was evaluated in the rice plants at 3, 6, 9 and 12 d after treatments by determining the chlorophyll level of the rice plants by the SPAD index using a portable chlorophyll meter, Minolta SPAD-502. Four readings were made per plant, taken over the center area of fully developed apical leaves, assigning the average value to each plant.

Phytosiderophore (PS) assessments were carried out at 6, 12, 24, 48 and 72 h after the treatment´s application, using the methodology described by Inal et al. [78] and Reichman et al. [79]. Rice plants were removed from the treatment and washed three times with deionized water. They were then placed in containers with 70 mL of deionized water for 3 h with constant aeration. From the root exudates produced in these 3 h, 9 mL aliquots were taken in vials, where 0.5 mL of 0.6 μM FeCl_3_ was added. They were stirred for 1 h to form the Fe (III)-PS compounds. Immediately afterwards, 1 mL of 1.0 M sodium acetate buffer (pH 7.0) was added to the solutions, and the mixture was stirred for 15 min to precipitate the remaining Fe (III). Next, the solutions were filtered through a 0.2 µm filter to remove any solid particles, and then 0.25 mL of 6 M HCL and 0.5 mL of 80 g L^−1^ hydroxylamine hydrochloride were added to reduce Fe (III) to Fe (II). The solutions were then placed in an oven and maintained at a temperature between 50–60 °C for 30 min. After the incubation, 0.25 mL of 2.5 g L^−1^ and 1 mL of 2.0 M sodium acetate buffer (pH 4.7) were introduced to the mixture. Finally, the contents of the tubes were mixed by shaking them briefly for 5 min.

The absorbance was determined at 562 nm. PS release rates were calculated as Fe equivalents.

After PS determination, roots were collected and kept at −80 °C for gene expression determination using primer pairs shown on Table 1.

### 4.5. Fusarium oxysporum Colonization Studies in Rice Roots

The GFP-tagged *F. oxysporum* FO12 transformant (GFP-FO12) was used to monitor the infection and colonization process of an entire rice plant growing in hydroponic system. Fifty plants were inoculated with the GFP-FO12, and the colonization of root tissue samples was visualized by CLSM (confocal laser scanner microscopy) over the 21 d post-inoculation period.

Two different treatments were set up in the presence of the GFP-FO12 to assess the effect of the concentration of Fe in the GFP-FO12 colonization process of root tissues. Thus, 25 inoculated plants were growth in the absence of Fe and 25 inoculated plants were grown with 70 µM Fe concentration. Two plants per treatment were analysed per day until 7 d after inoculation (7 dai). From this time point, two plants per treatment were analysed every 2 days until the end of the bioassay (21 dai). All the inoculated plants were examined by CLSM.

Rice tissue samples for microscopic studies were prepared according to the protocol previously described by Prieto et al. [79]. Rice roots were thin enough to perform CLSM analysis without vibratome sectioning. Therefore, whole roots were used to visualize the *F. oxysporum* colonization process. At least ten different roots per plant were mounted in a slice with distilled water to perform CLSM analysis.

Whole root tissues from the different treatments were used to collect single confocal optical sections using an Axioskop 2 MOT microscope (Carl Zeiss, Jena, Germany) equipped with a krypton and an argon laser, controlled by Carl Zeiss Laser Scanning System LSM5 PASCAL software (Carl Zeiss). GFP-FO12 was visualized using a 488 nm argon laser light (detection at 500–520 nm). Finally, data were recorded and the images relocated for analysis to Zeiss LSM Image Browser version 4.0 (Carl Zeiss). Confocal stacks were mounted and analysed to assess colonization of GFP-FO12. Images included in Figure 5 were obtained from projections of adjacent confocal optical sections. Final figures were handled with PhotoShop 10.0 software (Adobe Systems, San Jose, CA, USA).

### 4.6. qRT-PCR Analysis

Genes related to PS production by root cells of Fe-deficient rice plants were analysed. The following genes were analysed: *OsTOM1*, a deoxymugineic acid (DMA) efflux transporter; *OsNAAT*, which participates in DMA biosynthesis for Fe (III)-DMA uptake and translocation; *OsYSL15*, a Fe (III)-DMA transporter; *OsIRT1*, a Fe^2+^ transporter; and *OsIRO2*, which is an essential regulator involved in mediation of Fe uptake.

Roots were first ground into a fine powder using a mortar and pestle in liquid nitrogen. Total RNA was then extracted from the powdered roots using Tri Reagent solution (Molecular Research Center, Inc., Cincinnati, OH, USA), following the manufacturer’s instructions. To generate cDNA, 3 μg of DNase-treated root RNA was reverse transcribed using M-MLV reverse transcriptase (Promega, Madison, WI, USA) and random hexamers for amplification. For the study of gene expression, quantitative real-time polymerase chain reaction (qRT-PCR) was performed on a qRT-PCR Bio-Rad CFX connect thermal cycler. The amplification profile involved cycles with the following conditions: initial denaturation and polymerase activation at 95 °C for 3 min, followed by amplification and quantification at 90 °C for 10 s, 57 °C for 15 s, and 72 °C for 30 s. A final melting curve stage was performed from 65 to 95 °C with an increment of 0.5 °C for 5 s to ensure the absence of primer dimer or nonspecific amplification products. The PCR reactions were set up in 20 μL of SYBR Green Bio-RAD PCR Master Mix, following the manufacturer’s instructions. To detect any contamination in the reaction components, controls containing water instead of cDNA were included. For normalization of gene expression, a reference gene (*OsActin*) was used. The specific primers utilized in the qRT-PCR analysis are listed in Table 1.

### 4.7. Statistical Analysis

The statistical analyses were carried out using IBM SPSS Statistics 25. To assess the normal distribution of the variables studied, the Shapiro–Wilk normality test was applied. If the significance value was greater than 0.05, the data were considered to follow a normal distribution (parametric). Conversely, if the significance value was below 0.05, the data were considered non-parametric. For comparisons between the inoculated and control treatments, either Student’s *t*-test (parametric) or the Mann–Whitney test (non-parametric) was used to determine significant differences (*p* < 0.05). To compare gene expression between control and inoculated treatments at different times, analysis of variance (One-way ANOVA) and Dunnett’s test were utilized, setting the threshold for statistical significance at *p* < 0.05.

## 5. Conclusions

In conclusion, the results of this study demonstrate that the FO12 strain exhibits promising characteristics as a biofertilizer for rice plants. Its ability to colonize rice roots under Fe-deficient conditions, induce the expression of Fe-relative genes, increase phytosiderophore production and promote plant growth and development highlights its potential as an effective Fe biofertilizer. However, further research is necessary to fully understand its mechanisms of action and to optimize its application for agricultural purposes. Continued investigation and experimentation will be crucial in harnessing the full potential of the FO12 strain as a biofertilizer to enhance Fe uptake and improve crop productivity in rice and potentially other agricultural systems. Our results also indicate that the FO12 strain can indeed induce Fe-deficiency responses in rice plants, similarly to other ISR-eliciting microbes.

## Figures and Tables

**Figure 1 plants-12-03145-f001:**
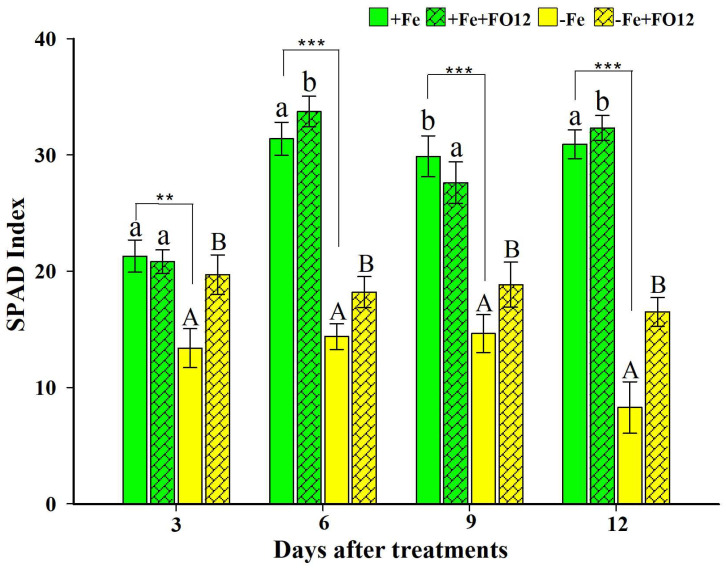
Effect of the inoculation with the nonpathogenic strain *Fusarium oxysporum* FO12 on the SPAD index of rice plants grown under Fe sufficiency or Fe deficiency. SPAD index determinations were carried out at 3, 6, 9 and 12 d after treatments. Treatments: –Fe, –Fe+FO12, +Fe and +Fe+FO12. The values represented are mean ± ES (*n* = 8). Within each time, *** p* < 0.01 or *** *p* < 0.001 indicate significant differences between treatments. For each evaluation moment, different lowercase or capital letters indicate significant differences between non-inoculated or inoculated plants with FO12 for +Fe or –Fe treatments, respectively.

**Figure 2 plants-12-03145-f002:**
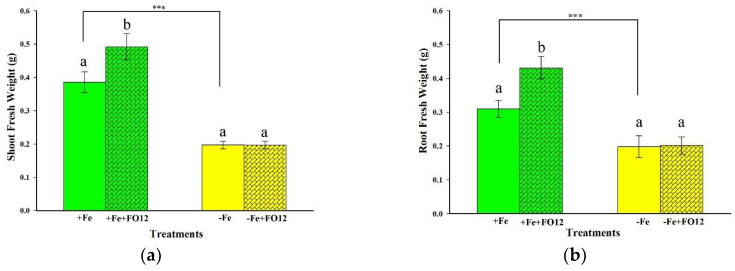
Effect of Fe deficiency and inoculation with FO12 in the growth of rice plants. (**a**) Shoot fresh weight. (**b**) Root fresh weight. To determine this effect, half of the 22 d old plants were inoculated. Then, both inoculated and control plants were cultivated for 12 additional days, either under Fe sufficiency (+Fe) or Fe deficiency (–Fe). After that time, roots and shoots were excised and weighed separately. The values represented are mean ± ES (*n* = 8). Different letters indicate significant differences according to Duncan’s multiple range test (*p* < 0.05). Similarly, *** *p* < 0.001 indicate significant differences between treatments.

**Figure 3 plants-12-03145-f003:**
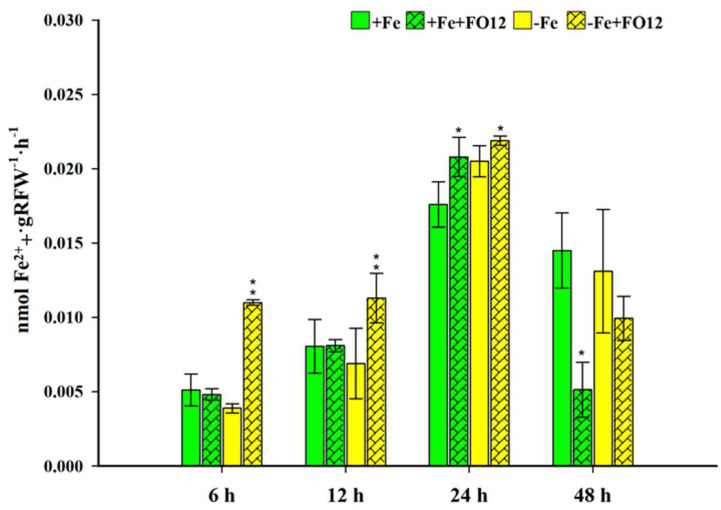
Evolution of phytosiderophore production in rice plants during 48 h of treatments. Four treatments were carried out: plants with Fe (+Fe), plants with Fe and inoculated with FO12 (+Fe+FO12), plants without Fe (–Fe), plants without Fe and inoculated with FO12 (–Fe+FO12). The inoculation was carried out the same day the Fe-deficiency treatment was applied. Within each sampling time, * or ** indicate significant differences (*p* < 0.05 or *p* < 0.01) relative to their respective non-inoculated control plants.

**Figure 4 plants-12-03145-f004:**
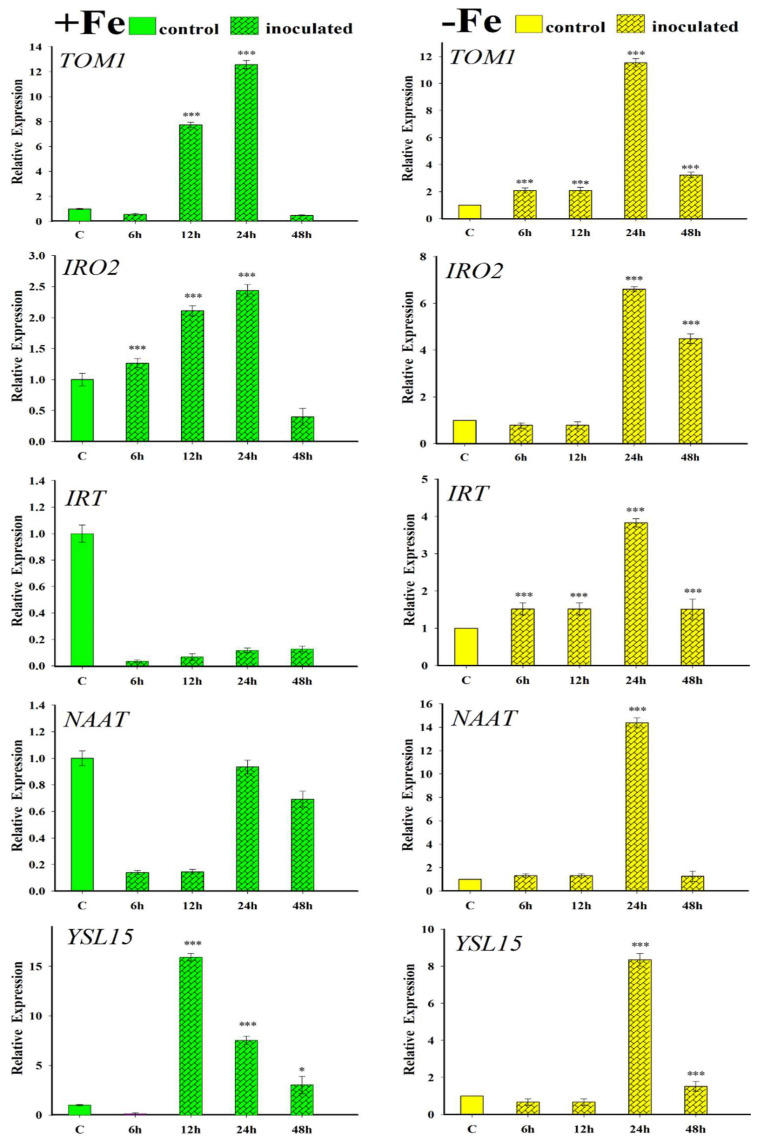
Effect of FO12 on the relative expression of PS-related genes (*TOM1*, *IRO2*, *IRT1*, *NAAT* and *YSL15*) in roots of rice plants. Four treatments were carried out: plants with Fe (+Fe), plants with Fe and inoculated with FO12 (–Fe+FO12), plants without Fe (–Fe), and plants without Fe and inoculated with FO12 (+Fe+FO12). The data represent the mean ± SE of three independent biological replicates and two technical replicates 2 d after treatments. Within each time, * *p* < 0.05 or *** *p* < 0.001 indicate significant differences in relation to the control treatment.

**Figure 5 plants-12-03145-f005:**
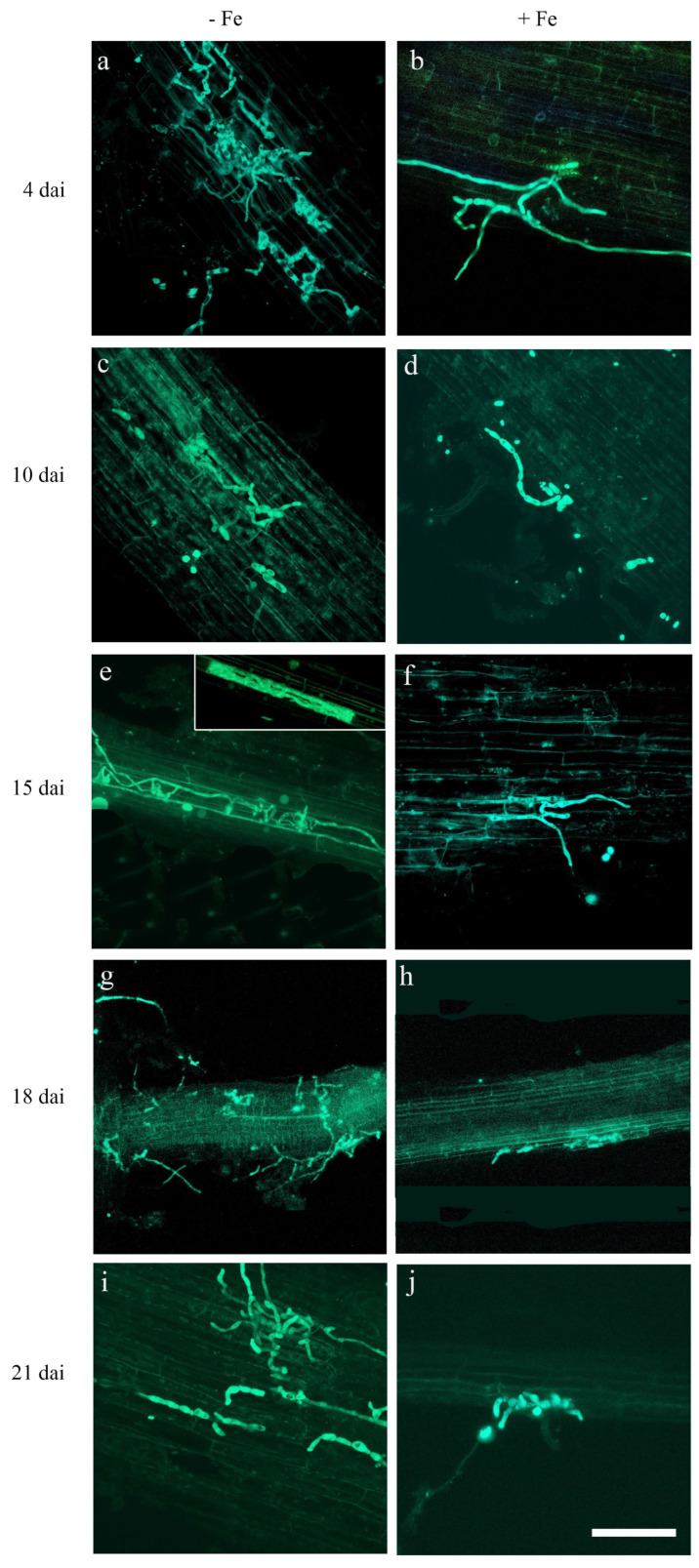
CLSM images of the time-course colonization processes of rice roots by the GFP-FO12 (in green). Confocal analysis was carried out on 4–5 cm long roots to show surface GFP-FO12 colonization. Images are projections of 20 adjacent confocal optical sections. The focal step between confocal optical sections was 0.5 μm. (**a**,**b**) Surface colonization at 4 dai by GFP-FO12 on rice roots of plants (**a**) without Fe, and with a supplement of (**b**) 70 μM Fe. (**c**,**d**) Surface colonization at 10 dai by GFP-FO12 on rice roots of plants growing (**c**) without Fe and (**d**) with an addition of 70 μM Fe. (**e**,**f**) Surface and internal (inset) colonization at 15 dai by GFP-FO12 on rice roots of plants (**e**) without Fe and (**f**) with a supplement of 70 μM Fe. (**g**,**h**) Surface colonization at 18 dai by GFP-FO12 on rice roots of plants (**g**) without Fe and (**h**) with a supplement of 70 μM Fe. (**i**,**j**) Surface colonization at 21 dai by GFP-FO12 on rice roots of plants (**i**) without Fe and (**j**) with a supplement of 70 μM Fe. Rice roots were colonized by GFP-FO12 hyphae in plants growing both in the presence and in the absence of Fe during the whole bioassay.

**Table 1 plants-12-03145-t001:** Primer pairs used for rice gene expression analysis.

Gene	Sequence 5-3
*OsNAAT1*	Forward: TAAGAG GATAATTGATTTGCTTAC
	Reverse: CTG ATCATTCCAATCCTAGTACAAT
*OsYSL15*	Forward: AACATAAGGGGGACTG GTAC
	Reverse: TGATTACCGCAATGATGCTTAG
*OsIRO2*	Forward: CTCCCATCGTTTCGGCTACCT
	Reverse: GCTGGGCACTCCTCGTTGATC
*OsTOM1*	Forward: GCCCAAGAACGCCAAAATGA
	Reverse: GGCTTGAAGGTCAACGCAAG
*OsIRT1*	Forward: CGTC TTCTTCTTCTCCACCACGAC
	Reverse: GCAGCTGATGATCGAGTCTG ACC
*OsActin*	Forward: TGCTATGTACGTCGC CATCCAG
	Reverse: AATGAGTAACCACGCTCCGTCA

## Data Availability

The original contributions presented in the study are included in the article.: All the data included in this article are publicly available. Further inquiries can be directed to the corresponding author.

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
