# Peer review of "Effect of the Nonpathogenic Strain Fusarium oxysporum FO12 on Fe Acquisition in Rice (Oryza sativa L.) Plants"

_plants, 2023, doi:10.3390/plants12173145_

Round 1
Reviewer 1 Report
Comments to the Author
The manuscript on “Effect of the nonpathogenic strain Fusarium oxysporum FO12 on Fe acquisition in rice (Oryza sativa L.) plants”. Authors have investigated the non-pathogenic strain FO12 on rice for Fe acquisition. The manuscript can be accepted for publication.
1. There is no information on the identity of the FO12 strain by genome sequencing. Authors should mention strain essential information.
2. Authors should substantiate FO12 strain is non-pathogenic by comparing it with a virulent strain. Experimental data needs to be included in the manuscript.
3. Line 197. FO1? Is that right?
4. Experiments were done in hydroponics in testing FO12 strain, but it would give more insight if conducted in the greenhouse using field soil.
Author Response
Reviewer 1
- There is no information on the identity of the FO12 strain by genome sequencing. Authors should mention strain essential information.
We sincerely appreciate your considerate feedback. The characterization of the non-pathogenic FO12 strain has been studied previously and it is well supported by the previous studies conducted first by Varo et al. 2016, and subsequently, more specifically, by Mulero-Aparicio et al. during his PhD Thesis, which was mainly focused on this topic. Thus, the authors consider that it is not necessary to provide additional information about the genome of FO12 in this paper since it is not the target task here, besides it is well supported by the previous studies.
According to Mulero-Aparicio et al. 2019a finally, it is important to discuss the non-pathogenic character of the FO12 strain, which has already been evaluated in pathogenicity tests for a wide range of hosts (i.e. cotton, olive, pepper, tomato, melon, eggplant, lettuce, peas, sunflower, etc.) inoculated with FO12. In all cases, no symptoms associated with Fusarium wilt were observed several weeks after inoculation (Mulero-Aparicio et al. unpublished results). In addition, previous studies have demonstrated that the non-pathogenic strains of F. oxysporum lack accessory chromosomes that regulate pathogenicity (Ma et al., 2010; van der Does et al., 2016; van Dam et al., 2017). Therefore, molecular characterization of the FO12 strain needs to be conducted to determine whether this strain contains packages of pathogenicity genes in its genome and to consequently discard the possibility that this isolate becomes pathogenic over time.
To clarify this statement, we have integrated a new paragraph in the justification and objective section in the introduction, including all the following references:
- Mulero-Aparicio, A., Agustí Brisach, C., Varo, A., López-Escudero, F.J., Trapero, A., 2019a. A non-pathogenic strain of Fusarium oxysporum as a potential biocontrol agent against Verticillium wilt of olive. Biological Control, 139: Article 104045
- Mulero-Aparicio A, Cernava T, Turrà D, Schaefer A, Di Pietro A, López-Escudero FJ, Trapero A and Berg G 2019b. The Role of Volatile Organic Compounds and Rhizosphere Competence in Mode of Action of the Non-pathogenic Fusarium oxysporum FO12 Toward Verticillium Wilt. Microbiol. 10:1808. doi: 10.3389/fmicb.2019.01808
- Mulero-Aparicio, A. Trapero, F.J. López-Escudero 2020a. A nonpathogenic strain of Fusarium oxysporum and grape marc compost control Verticillium wilt of olive. Phytopathologia Mediterranea 59(1): 159-167. doi: 10.14601/Phyto-11106.
- Mulero-Aparicio, A., Varo, A., Agustí-Brisach, C., López-Escudero, F.J., Trapero, A., 2020b. Biological control of Verticillium wilt of olive in the field. Crop Protection, 128 Article 104993
- Authors should substantiate FO12 strain is non-pathogenic by comparing it with a virulent strain. Experimental data needs to be included in the manuscript.
We are truly grateful that you've highlighted this matter. Your feedback holds significant value for us. It is essential to establish the justification for the focus of this work, which is centered around clarifying the role of FO12 as an inducer of resistance and its involvement in iron acquisition. It's important to note that this work is distinct from a disease biocontrol study, rendering the comparison irrelevant. This line of reasoning is reinforced by our prior investigations involving FO12 within this context, as well as by findings from other researchers working with various non-pathogenic FO strains that were not subjected to comparisons with pathogenic counterparts.
- Line 197. FO1? Is that right?
Thank you for identifying that typo. It has already been rectified in the latest version of the manuscript.
- Experiments were done in hydroponics in testing FO12 strain, but it would give more insight if conducted in the greenhouse using field soil.
This study represents a preliminary exploration focused on elucidating the impact of the FO12 strain on ferric nutrition of rice plants. Its primary objective has been to investigate the strain's influence on phytosiderophore production or the relative expression of PS-related genes in rice root systems, among other factors. The hydroponic system was chosen as the optimal method for conducting these assessments. We acknowledge your perspective on the matter, and we agree that conducting similar experiments in field soil would provide a broader perspective. In fact, our research group is already doing a series of experiments involving rice plant cultivation in pots containing calcareous soil within a greenhouse environment. The goal of these experiments is to comprehensively evaluate the effects of the FO12 strain and other beneficial microorganisms on growth and production outcomes. We anticipate that the outcomes of these experiments will be presented in forthcoming publications.
Reviewer 2 Report
The manuscript titled “Effect of the nonpathogenic strain Fusarium oxysporum FO12 on Fe acquisition in rice (Oryza sativa L.) plants” by Jorge Núñez-Cano et al. builds upon authors previously published research on cucumber and nonpathogenic strain F. oxysporum FO12.
Overall, the manuscript describes comprehensive research work and provides methodology for determination of the ability of the nonpathogenic strain Fusarium oxysporum FO12 to induce Fe deficiency responses in rice plants and its effects on plant growth, Fe chlorosis, phytosiderophores production, expression of Fe-related genes, and FO12 colonization in rice root tissues under hydroponic system conditions.
Page 3, Line 126 -129: However, the claim that nonpathogenic strains of Fusarium oxysporum induce this resistance (EMR) is a subject of debate due to conflicting evidence. For instance, research has shown that when Capsicum annuum plants are inoculated with the nonpathogenic strain of F. oxysporum FO47, they gain resistance against Verticillium dahlia and experience reduced foliar damage [48].
Here, the authors argument (based on the cited work) is incomplete, does not provide all the relevant information. If the idea is to differentiate EMR effect based on the involvement or non-involvement of phytohormones (ethylene) in host plant protection against pathogens, it needs to be made clearer.
Page 11, Line 360 - 361: Collectively, these findings suggest that the depletion of Fe in the plant increases its vulnerability to pathogenic attacks from Fusarium races, while also promoting endophytic colonization by non-pathogenic strains, such as FO2.
Pathogenic Fusarium races were not evaluated in this study.
Author Response
Reviewer 2
- Page 3, Line 126 -129: However, the claim that nonpathogenic strains of Fusarium oxysporum induce this resistance (EMR) is a subject of debate due to conflicting evidence. For instance, research has shown that when Capsicum annuum plants are inoculated with the nonpathogenic strain of F. oxysporum FO47, they gain resistance against Verticillium dahlia and experience reduced foliar damage [48].
Here, the authors argument (based on the cited work) is incomplete, does not provide all the relevant information. If the idea is to differentiate EMR effect based on the involvement or non-involvement of phytohormones (ethylene) in host plant protection against pathogens, it needs to be made clearer.
Thank you for your thoughtful feedback. In alignment with your observation, we recognize the importance of refining our argument. In this context, we have incorporated the following paragraph to enhance clarity:
'Constantin et al. (2019) have proposed that endophytic microorganisms can induce EMR, a resistance mechanism distinct from ISR, with the intriguing feature of ethylene-independence. This assertion implies that the introduction of nonpathogenic strains of F. oxysporum to plant hosts would activate resistance pathways not reliant on ethylene signaling. However, our own research, documented and published by our group, has revealed a contrary finding. Specifically, the inoculation of plants with the FO12 strain resulted in a noticeable augmentation of ethylene-related gene expression at specific intervals (Aparicio et al. 2023).'
We appreciate your input and believe that this addition will bolster the clarity and depth of our argument.
- Page 11, Line 360 - 361: Collectively, these findings suggest that the depletion of Fe in the plant increases its vulnerability to pathogenic attacks from Fusarium races, while also promoting endophytic colonization by non-pathogenic strains, such as FO2.
Pathogenic Fusarium races were not evaluated in this study.
Thank you for bringing this to our attention. Your feedback is greatly appreciated. We have duly taken your observation into consideration, and as a result, the sentence "increases its vulnerability to pathogenic attacks from Fusarium races" has been successfully removed from the text.
Reviewer 3 Report
The study titled " Effect of the nonpathogenic strain Fusarium oxysporum FO12 on Fe acquisition in rice (Oryza sativa L.) plants" examines the impact of the nonpathogenic strain Fusarium oxysporum FO12 on rice plants. The goal is to determine if the strain can cause iron deficiency in the plants and how it affects their growth and chlorosis.
General note:
The subject of the study is interesting and topical, with scientific and practical importance.
The introduction is presented correctly, in accordance with the subject. Numerous scientific articles, in concordance with the topic of the study, were consulted.
The study methodology was clearly presented, and appropriate to the proposed objectives.
The obtained results have been analyzed and interpreted correctly in accordance with the current methodology.
The discussions are appropriate in the context of the results and were conducted compared to other studies in the field.
The scientific literature to which the reporting was made is recent and representative of the field.
There are some minor changes I am suggesting in the detailed comments below.
Line 50: “liker” - Please check and improve.
Line 65 - Please check and improve the format.
Line 103 - Please check and improve the format.
Line 132 - Please check and improve the format.
Lines 137 – 139 I believe that the introduction section does not require the inclusion of the research's results. It would be more appropriate to place this statement in the Conclusion section. Kindly consider removing this sentence.
Lnes 144-149 – “In this study, Fe chlorosis was evaluated in rice plants at 3, 6, 9 and 12 d after treatments by determining the SPAD index by a portable chlorophyll meter Minolta SPAD-502. Four readings were made per plant, taken over the area center of fully developed apical leaves, assigning the average value to each plant.”. - It would be more appropriate to place this statement in the Materials and Methods section.
Figure 1 - the method of presenting significant differences between the objects is misleading (e.g. on the 12th day +Fe and -Fe objects were combined as significantly different at the level of ***P < 0.001, meanwhile small letters "a" were used, suggesting no differences). This should be corrected.
Line 243- the use of "we" should be avoided in articles
Lines 286-289 – “The main symptom of Fe deficiency in plants is an interveinal yellowing of the youngest leaves known as Fe chlorosis. It is produced because Fe plays an important role in the functioning of some enzymes involved in the synthesis of chlorophyll [23,49].” repeating. These sentences have already appeared in the Result section.
Lines 320, 327 - Please check and improve the format.
Line 429 – space is missing
Authors should revise the reference list by following the rules described in the guidelines for authors.
Author Response
Reviewer 3
Line 50: “liker” - Please check and improve.
Thank you for identifying that typo. It has already been rectified in the latest version of the manuscript.
Line 65 - Please check and improve the format.
Thank you for identifying that typo. It has already been rectified in the latest version of the manuscript.
Line 103 - Please check and improve the format.
Thank you for identifying that typo. It has already been rectified in the latest version of the manuscript.
Line 132 - Please check and improve the format.
Thank you for identifying that typo. It has already been rectified in the latest version of the manuscript.
Lines 137 – 139 I believe that the introduction section does not require the inclusion of the research's results. It would be more appropriate to place this statement in the Conclusion section. Kindly consider removing this sentence.
Thank you for your collaboration in refining the text. The sentence "Our results indicate that the FO12 strain can indeed induce Fe deficiency responses in rice plants, similar to other ISR-eliciting microbes." has been successfully removed from the Introduction to Conclusions, as per your suggestion.
Lnes 144-149 – “In this study, Fe chlorosis was evaluated in rice plants at 3, 6, 9 and 12 d after treatments by determining the SPAD index by a portable chlorophyll meter Minolta SPAD-502. Four readings were made per plant, taken over the area center of fully developed apical leaves, assigning the average value to each plant.”. - It would be more appropriate to place this statement in the Materials and Methods section.
Thank you, again, for your collaboration in refining the text. The sentence "In this study, Fe chlorosis was evaluated in rice plants at 3, 6, 9 and 12d after treatments by determining the SPAD index by a portable chlorophyll meter Minolta SPAD-502. Four readings were made per plant, taken over the area center of fully developed apical leaves, assigning the average value to each plant." has been successfully removed from the Results to Materials and Methods section, as per your suggestion.
Figure 1 - the method of presenting significant differences between the objects is misleading (e.g. on the 12th day +Fe and -Fe objects were combined as significantly different at the level of ***P < 0.001, meanwhile small letters "a" were used, suggesting no differences). This should be corrected.
Thank you for your correction since it has improved the interpretation of the Figure 1 for the readers. To avoid misunderstandings, we have used lowercase or capital letters to indicate significant difference between inoculated or non inoculated plants within the Fe+ or Fe- treatments, respectively. We have also included this explanation in the figure caption.
Line 243- the use of "we" should be avoided in articles
The pronoun "We" has been substituted for "It" in accordance with your recommendation. Thank you for your guidance in enhancing the text.
Lines 286-289 – “The main symptom of Fe deficiency in plants is an interveinal yellowing of the youngest leaves known as Fe chlorosis. It is produced because Fe plays an important role in the functioning of some enzymes involved in the synthesis of chlorophyll [23,49].” repeating. These sentences have already appeared in the Result section.
Thank you for your guidance in enhancing the text. The sentence that you propose has been modified by this other: “The primary indication of Fe deficiency in plants manifests as interveinal yellowing in the most juvenile leaves, a condition referred to as Fe chlorosis. This discoloration arises due to iron's pivotal function within several enzymes engaged in chlorophyll synthesis [23,49].”
Lines 320, 327 - Please check and improve the format.
We apologize for any inconvenience, but we were unable to locate any typos on lines suggested. Could you kindly specify the areas that require improvement within these lines? Your guidance would be greatly appreciated.
Line 429 – space is missing
Thank you for identifying that typo. It has already been rectified in the latest version of the manuscript.
Authors should revise the reference list by following the rules described in the guidelines for authors.
We extend our gratitude for your invaluable assistance in refining the formatting. We have successfully incorporated new references into the list and made the necessary corrections, aligning them meticulously with the guidelines outlined for authors. Your guidance has been very appreciated.